

# Mayfly emergence production and body length response to hydrology in a tropical lowland stream

Pablo E. Gutiérrez-Fonseca[1] and Alonso Ramírez[2]

[1] Department of Biology & Center for Research in Marine Science and Limnology, University of Costa Rica, San José, Costa Rica
[2] Department of Applied Ecology, North Carolina State University, Raleigh, NC, USA

## ABSTRACT

**Background:** Hydrological impacts on aquatic biota have been assessed in numerous empirical studies. Aquatic insects are severely affected by population declines and consequent diversity loss. However, many uncertainties remain regarding the effects of hydrology on insect production and the consequences of energy transfer to the terrestrial ecosystem. Likewise, sublethal effects on insect morphology remain poorly quantified in highly variable environments. Here, we characterized monthly fluctuation in benthic and emerged biomass of Ephemeroptera in a tropical lowland stream. We quantified the proportion of mayfly production that emerges into the riparian forest. We also examined the potential morphological changes in *Farrodes caribbianus* (the most abundant mayfly in our samples) due to environmental stress.
**Methods:** We collected mayflies (nymphs and adults) in a first-order stream in Costa Rica. We compared benthic and adult biomass from two years' worth of samples, collected with a core sampler (0.006 m$^2$) and a 2 m$^2$-emergence trap. The relationship between emergence and annual secondary production (E/P) was used to estimate the Ephemeroptera production that emerged as adults. A model selection approach was used to determine the relationship between environmental variables that were collected monthly and the emergent biomass. To determine potential departures from perfect bilateral symmetry, we evaluated the symmetry of two morphological traits (forceps and forewing) of *F. caribbianus* adults. We used Spearman's rank correlation coefficients ($\rho$) to examine potential changes in adult body length as a possible response to environmental stress.
**Results:** Benthic biomass was variable, with peaks throughout the study period. However, peaks in benthic biomass did not lead to increases in mayfly emergence, which remained stable over time. Relatively constant mayfly emergence suggests that they were aseasonal in tropical lowland streams. Our E/P estimate indicated that approximately 39% and 20% (for 2002 and 2003, respectively) of the nymph production emerged as adults. Our estimated proportion of mayfly production transferred to terrestrial ecosystems was high relative to reports from temperate regions. We observed a strong negative response of *F. caribbianus* body length to increased hydrology (Spearman: $\rho = -0.51$, $p < 0.001$), while slight departures from perfect symmetry were observed in all traits.
**Conclusion:** Our two years study demonstrates that there was large temporal variability in mayfly biomass that was unrelated to hydrological fluctuations, but potentially related to trophic interactions (e.g., fish predation). Body length was a

Corresponding author
Pablo E. Gutiérrez-Fonseca,
pabloe.gutierrezfonseca@gmail.com

good indicator of environmental stress, which could have severe associated costs for mayfly fitness in ecosystems with high temporal variation. Our results highlight the complex ecological and evolutionary dynamics of tropical aquatic insects, and the intricate connection between aquatic and terrestrial ecosystems.

## INTRODUCTION

Identifying factors that drive changes in natural communities has been a key issue in ecology, because it allows us to understand current patterns and to predict community responses to environmental change (*Power et al., 1988*; *Resh et al., 1988*). In freshwater ecosystems, much attention is given to understand the impact of environmental variables on aquatic organisms at multiple levels, from individuals to the community (*Ardón et al., 2013*; *Klem & Gutiérrez-Fonseca, 2017*). Among a wide range of factors, hydrology has often been reported as most prominent affecting aquatic biota (*Ramírez & Pringle, 1998*; *Ríos-Touma, Encalada & Prat Fornells, 2011*). Thus, while it is well known that hydrology reduces populations by catastrophic mortality, channel scouring and resource redistribution; we know less about sublethal stresses that elicit escape (e.g., early emergence) or generate morphological changes.

Insect emergence can be used as a reliable indicator of population success as it reflects the influence of multiple environmental stressors that populations face during their larval development. Temporal patterns of insect emergence are often synchronized and occur during a limited period of time (*Castro-Rebolledo & Donato-Rondon, 2015*). However, hydrology may strongly affect seasonality, magnitude and timing of emergence (*Whiles & Goldowitz, 2001*; *Lytle, 2002*). This can have negative effects in aquatic-terrestrial energy fluxes, since emerged aquatic insects provide significant subsidies for riparian food webs (*Nakano & Murakami, 2001*). Thus, although emergent biomass may be a small fraction of benthic biomass (*Statzner & Resh, 1993*), changes in the number and morphology (e.g., body length) of individuals can have a large impact on the biomass and nutrient export to adjacent terrestrial ecosystem (*Small et al., 2013a*; *Kelly, Cuevas & Ramírez, 2015*).

Individual body length and deviation from perfect bilateral symmetry are particularly useful as measures of developmental stability (i.e., ability of an individual to adequately buffer its developmental processes against environmental and genetic perturbations and ensure common developmental outcomes under particular ecological and evolutionary conditions). Environmental stressors affect body length by affecting growth or development rates, which carries significant consequences for individual fitness and alters mortality rates and reproductive success (*Peckarsky et al., 2001*; *Dahl & Peckarsky, 2003*). Bilateral symmetry is known for many traits, and departure from symmetrical phenotypes has been linked to low success of individuals in sexual competition (*Møller, 1990*; *Santos, 2001*; *Jorge & Lomônaco, 2011*). Departures from symmetry are commonly grouped into three categories, based on frequency distributions of the differences between the right

and left sides of a structure: (a) directional asymmetry (i.e., greater development of a character on one side), (b) antisymmetry (i.e., asymmetry without directional bias) and (c) fluctuating asymmetry (i.e., random departure from perfect symmetry of any bilateral anatomical character, showing a normal distribution with a mean of zero). The first two cases of asymmetry are related to heritable variation for asymmetry, while fluctuating asymmetry is commonly caused by environmental disturbance (*Palmer & Strobeck, 1986*, *1992*).

Tropical streams are highly variable in their environmental parameters, which has an influence on aquatic biota (*Jacobsen & Encalada, 1998*; *Ramírez, Pringle & Douglas, 2006*). Streams draining humid tropical rainforests often experience unpredictable hydrological events, which may represent sources of stress to aquatic populations. This is especially true for tropical mayflies, which may live relatively long-periods in the streams (range from 26 to 165 days, *Sweeney, Jackson & Funk, 1995*) compared to their lifespan as adults (range from 3 to 6 days, *Vásquez, Flowers & Springer, 2009*). Streams at La Selva Biological Station (LSBS) offer an excellent opportunity to assess how environmental stressors influence aquatic biota, as they show high interannual variability in their environmental variables (*Ramírez, Pringle & Douglas, 2006*; *Small et al., 2012*; *Gutiérrez-Fonseca, Ramírez & Pringle, 2018*).

In this study, we examine the benthic and emerged biomass of Ephemeroptera in a small tropical lowland stream. We quantified the biomass transfer to the terrestrial ecosystem by adult emergence production (i.e., E/P ratio). We also assess whether environmental variability can influence mayfly morphology (i.e., symmetry and body length). We approach our objectives in four ways: first, we used data collected during two years (2002–2003) to determine temporal patterns of emerged Ephemeroptera and the benthic standing-stock biomass. Then, we estimated the E/P ratio using adult biomass and nymph secondary production for each year. Second, we identified which environmental variables were related to the emergence patterns of mayflies. Third, we examined departures from perfect symmetry in adults of the Leptophlebiidae *Farrodes caribbianus* (Traver) comb. nov. (*Domínguez, 1999*), the most abundant Ephemeroptera found in our emergence trap. Fourth, we assessed potential changes in body length of *F. caribbianus* and how they relate to the variation in rainfall. We focus on precipitation as a key factor impacting macroinvertebrates, since previous studies have demonstrated their influence on LSBS streams (*Ramírez, Pringle & Douglas, 2006*; *Gutiérrez-Fonseca, Ramírez & Pringle, 2018*). We hypothesized large mayfly mortality due to drag forces experienced by individuals during floods, as well as an increase in nymph development instability due to exposure to environmental stressors during their lifespan. We expected to observe a peak in benthic and emergent biomass during the low rainfall season, as well as changes in adult symmetry and body length.

## MATERIALS AND METHODS

### Study system

This study was conducted at LSBS (10°26′ N, 84°01′ W), a 1,563 ha reserve in the Caribbean slope of Costa Rica, located in a gradient break between the Cordillera

Central and the coastal plain. The forest in LSBS is composed of mature and secondary tropical rainforest (*Holdridge, 1967*). Long-term average annual precipitation (1963–2016) is 4,354 mm, ranging from 2,809 mm in 1995 to 6,165 mm in 1970 (available at http://www.ots.ac.cr/meteoro/). The annual distribution is bimodal, with peaks of >400 mm/mo occurring both in June–July and November–December. The period with low rainfall values is February–April (*Sanford et al., 1994*).

We collected the Ephemeroptera samples (nymphs and adults) from Carapa, a first order stream (1 m wide and 0.25 m deep) bordered with abundant riparian vegetation (canopy cover: 85%, *Small et al., 2013b*). We obtained the samples from an approximately 100 m reach, which was representative of overall stream conditions. Within the study reach, channel substrate type was consistently dominated by detritus and fine sediments (i.e., silt and clay). Long-term data sets (1997–2011) show that discharge ranges from 0.011 to 0.027 $m^3$/s, stream temperature from 21.4 to 27.2 °C and pH from 3.62 to 6.46, with low values occurring during the El Niño event of 1997–1998 (*Small et al., 2012*; *Gutiérrez-Fonseca, Ramírez & Pringle, 2018*).

Benthic macroinvertebrate assemblages in Carapa are diverse, and include several species of dipterans, mayflies, caddisflies, odonates, beetles and non-insects. Diptera dominates the taxonomic richness, abundance and biomass of insects. Odonata, Trichoptera and Ephemeroptera are also numerically important groups (*Ramírez, Pringle & Douglas, 2006*; *Gutiérrez-Fonseca, Ramírez & Pringle, 2018*). Fish assemblages are dominated by the insectivorous fish *Priapicthys annectens* (98% of abundance, Family Poeciliidae) (*Small et al., 2013b*).

## Nymph biomass and production

We used data of mayfly benthic standing-stock biomass from *Gutiérrez-Fonseca, Ramírez & Pringle (2018)*. Ephemeroptera nymphs were sampled monthly for two years (2002–2003). Three core samples (0.006 $m^2$ each) were collected in runs with leaves as the dominant substrate. All material enclosed into the core sampler was removed to a depth of ~10 cm or until reaching bedrock. Mayfly nymphs were removed from organic matter and preserved in 80% ethanol. Biomass of individual nymphs was estimated by applying the length-mass relationship developed by *Benke et al. (1999)*, expressed as ash-free dry mass (AFDM) per area ($m^2$).

We were unable to calculate directly the annual secondary production (P) using samples collected in our study, as we did not find nymphs of all size classes on all sampling dates. Instead, we estimated P using the production-to-biomass (P/B) method (*Benke, 1993*). An average P/B value for Ephemeroptera at La Selva (13.63 $y^{-1}$) was obtained from *Ramírez & Pringle (1998)*, and biomass (B) was from our benthic sampling. While this is an indirect method, it should provide us with the best approximation given our limited information on tropical Ephemeroptera.

## Adult emergence and biomass

We used a 2 $m^2$ (sampling area) emergence trap (BioQuip Products, Rancho Dominguez, CA, USA) to sample mayfly adults continuously from July 2001 to February 2004. The trap

was suspended over the stream and covered the entire stream width, which allowed us to sample in various microhabitats such as riffles, pools, runs and stream banks. Emerging insects were collected weekly and preserved in 80% ethanol for subsequent taxonomic identification. A modified handheld vacuum was used to remove emergent insects from the trap. The trap was inspected often for maintenance (i.e., repair of holes and removal of spider webs). Mayfly biomass was calculated by measuring the length of each individual and applying the length–mass relationship developed by *Sabo, Bastow & Power (2002)*. Emergence biomass was expressed as mg AFDM/m$^2$ by taking the total biomass of each month adjusted by the trap area. Total annual biomass was determined by adding all weekly samples for each year.

## Physicochemical variables and hydrology

We measured eight physicochemical variables monthly, simultaneously with Ephemeroptera collections. Nutrient concentrations (i.e., $NO_3^-$-N, $NH_4^+$-N and $PO_4^{3-}$-P as soluble reactive P: SRP) were measured by collecting two filtered (0.45 μm Millipore filters) water samples using new 125 mL bottles. Samples were kept frozen until analyzed. $NO_3^-$-N, $NH_4^+$-N and SRP concentrations were measured using continuous-flow colorimetry and an Alpkem RFA 300 colorimetric analyzer. We used cadmium-reduction, phenol-hypochlorite and ascorbic acid methods for $NO_3^-$-N, $NH_4^+$-N and SRP, respectively (*APHA, 1998*). We measured stream temperature, pH, and conductivity in situ with a handheld meter (Hanna Instruments, Woonsocket, RI, USA). The stream flow was measured with a Marsh–McBirney current meter (Marsh McBirney Inc., Frederick, MD, USA), and discharge was calculated using the velocity–area method (*Gordon et al., 2004*). Monthly precipitation was recorded using data from the meteorological station available at LSBS (OTS meteorological data, http://www.ots.ac.cr/meteoro/).

## Measurement of traits and body length of *F. caribbianus*

We measured the body length (not including cerci) of all individual mayflies to quantify size variation patterns. To determine departures from symmetry, we measured the length of the second segment of forceps (SF) on the right and left sides of males, as well as forewing area (AFW) and forewing length (LFW) on the right and left sides of both males and females. Body parts were removed with forceps, mounted on glass slides, and photographed with a stereomicroscope (AmScope, Irvine, CA, USA) and a microscope (Nikon Eclipse E400). Images were analyzed with the free computer software ImageTool 2.0 (University of Texas Health Science Center, San Antonio, TX, USA). All linear and area measurements were done with an accuracy of 0.01 mm and 0.01 mm$^2$, respectively. To avoid mistaking human error with potential asymmetries, we used one image per individual to take three non-consecutive (i.e., in a random order) measurement of each trait.

## Data analyses

We used biomass to compare temporal patterns between benthic and emerging mayflies, since biomass better reflects potential changes in production driven by environmental

variability, beyond other metrics such as abundance and diversity (*Malison, Benjamin & Baxter, 2010*). Likewise, biomass estimates are widely used in analyses of food webs and secondary production. To estimate the fraction of Ephemeroptera production that emerged as an adult and was exported to riparian ecosystems, we calculated the ratio of adult emergence to nymph secondary production (E/P) for each year.

A model selection approach based on Akaike's Information Criteria (AIC, *Akaike, 1973*) was used to identify the best-fit model that included the environmental variables influencing adult mayfly biomass. Model constructions and selections are described in more detail in *Gutiérrez-Fonseca, Ramírez & Pringle (2018)*. Briefly, we build linear regression models using a forward selection procedure. We identified the model with the best quality (i.e., with a minimum number of required explanatory variables) based on the lowest AICc (<2Δ). Model averaging was used to draw conclusions when more than one model was included in the subset. Multicollinearity among variables was assessed by calculating the variance inflation factor (VIF). Environmental variables with a VIF > 10 were identified and removed from the analysis (*O'Brien, 2007*). Before building the models, we also excluded variables that were highly correlated ($|r| > 0.60$).

To examine the association between body length and hydrology, we first related monthly average precipitation to discharge using Pearson correlation coefficients. Then, we determined the relationship between body length and average precipitation in the 159 days before collecting the emergence trap using non-parametric Spearman's rank correlation coefficients, due to the non-normal distribution of the data. Spearman's rank correlations were calculated separately for males ($n = 67$), females ($n = 50$) and both sexes combined ($n = 117$). We used average precipitation of the 159 days, as this timeframe coincides with the life cycle of *Thraulodes* sp. (same family, Leptophlebiidae), which was estimated as the median days since an egg hatches until the individual reaches adulthood (*Jackson & Sweeney, 1995*).

Analyses of potential departure from symmetry were performed on approximately 30 randomly selected individuals for each test, following the procedure recommended by *Palmer & Strobeck (1986*, *2003)* and *Palmer (1994)*. We used the signed differences between right and left (R–L) to distinguish fluctuating asymmetry (must be normally distributed with zero mean) from antisymmetry and directional symmetry. The Shapiro–Wilks test was used to determine whether the data were normally distributed, which would rule out antisymmetry. Directional asymmetry is characterized by a normal distribution with a mean other than zero. Therefore, we also conducted one-sample *t*-tests to determine if the mean of signed (R–L) differed statistically from zero.

Analysis of potential fluctuating asymmetries were conducted for each trait using three indices recommended by *Palmer & Strobeck (2003)*: FA1, FA4a and FA10a. FA1, calculated as mean of |R–L|, is the recommended index because it is easy to understand. FA4a (0.798 $\sqrt{\text{var(R–L)}}$) has higher statistical power and represents the contribution of FA measurement error. The FA10a index describes the average difference between sides after measurement error has been partitioned out. This index was calculated using two-way mixed model ANOVAs with side (Fixed), individual (Random) and their interaction (Side × Individual). Then, we used the mean square (ME) to calculate FA10a
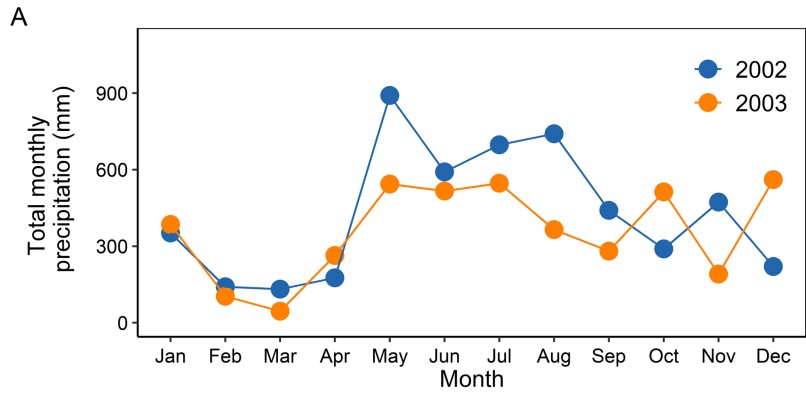

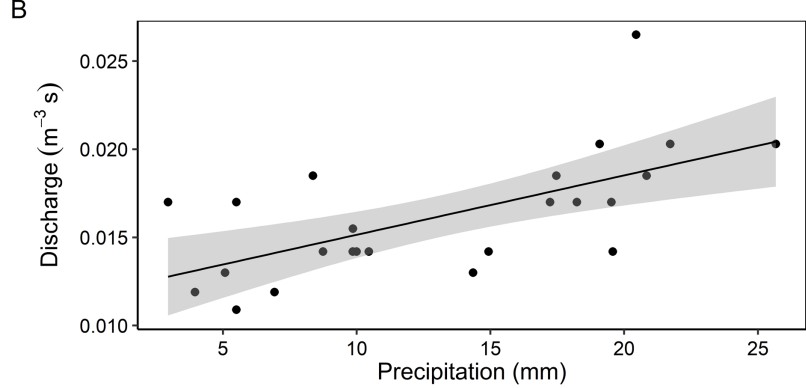

**Figure 1 Temporal variability of precipitation in 2002 and 2003 (A), and relationship between mean monthly precipitation and discharge (B).** Grey shaded area represents the 95% confidence intervals.

as $0.798 \sqrt{2\sigma_i^2}$ (where $\sigma_i^2 = ME_{SxI} - ME_{error}$). Additionally, we calculated measurement error as a percentage (ME3, according to *Palmer & Strobeck, 2003*) by $ME_{error}/ME_{SxI} \times 100$. ME3 represents the mean difference between replicate measurements as a proportion of mean difference between of the sides × individual interaction.

Pearson ($r$) and Spearman's rank ($\rho$) correlations were calculated with the *cor.test()* function of the *stats* package, the two-way mixed model ANOVA with the *lmer()* function of the *lme4* package (*Bates et al., 2015*), model averaging with the *AICcmodavg* package (*Mazerolle, 2019*) and graphics were produced using the *ggplot2* package (*Wickham, 2016*) in R version 3.6.3 (*R Core Team, 2019*). Raw data and code used in this study are available on a GitHub repository: https://github.com/PEGutierrezF/mayfly_morphometry.

## RESULTS

### Precipitation and hydrology

Monthly total precipitation showed a strong seasonal pattern during the study period. The dry season had the lowest precipitations in March 2002 (132.0 mm) and March 2003 (45.3 mm). The wet season had maximum values in May 2002 (890.8 mm) and December 2003 (561.3 mm) (Fig. 1A). Discharge reflected the monthly average precipitation variability in the two years of sampling (Pearson: $r = 0.64$, $p < 0.001$, Fig. 1B).

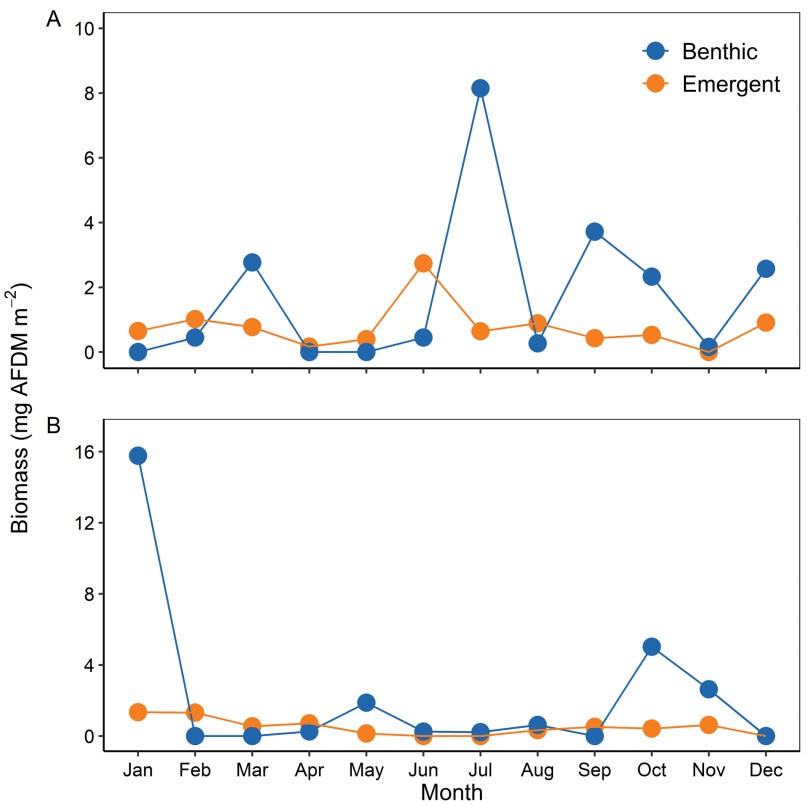

**Figure 2 Temporal variability of benthic and emerging adult biomass during (A) 2002 and (B) 2003.**

## Benthic and emergence biomass and E/P ratios

Mean annual biomass of nymphs was 1.74 mg AFDM/m$^2$ in 2002 and 2.22 mg AFDM/m$^2$ in 2003. Benthic biomass peaked in March, July, and September 2002 and January and October 2003 (Fig. 2). Meanwhile, mean annual biomass of emerging adults was 0.76 mg AFDM/m$^2$ in 2002 and 0.50 mg AFDM/m$^2$ in 2003. Monthly mayfly emergence was relatively constant throughout the study periods compared to benthic biomass, except for a slight increase in June 2002 (Fig. 2A). Total emergence was 9.14 mg AFDM m$^{-2}$ y$^{-1}$ and 5.99 mg AFDM m$^{-2}$ y$^{-1}$, while annual secondary production was 23.70 mg AFDM m$^{-2}$ y$^{-1}$ and 30.28 mg AFDM m$^{-2}$ y$^{-1}$ for 2002 and 2003, respectively. Therefore, we estimated that total emergence production represented 38.57% and 19.78% of nymph secondary production for 2002 and 2003, respectively.

## Physicochemical characteristics and individual-level variation

The AIC analysis used to determine the relative importance of environmental variables on the biomass of emerging adults showed no support for any model. Therefore, none of the variables related to nutrients, hydrology or physicochemistry that were evaluated in our study explained the relatively constant patterns of mayfly emergence in our streams.

Body length of *F. caribbianus* was strongly influenced by average rainfall during the 159 days of nymph development. Spearman's rank correlations revealed a negative

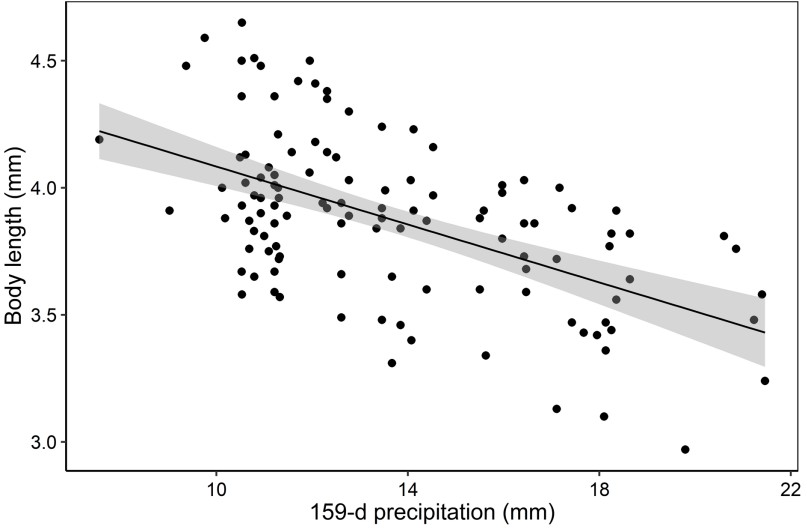

**Figure 3 Body length variability of *F. caribbianus* and precipitation.** Relationship between *F. caribbianus* body length and average precipitation in the 159 days prior to the sampling date. Each point represents an individual, including males and females. Gray shaded area represents the 95% confidence intervals. 

**Table 1 Test for normality and *t*-test for each trait.**

| Trait | N | Mean (R–L) | SE | Normality test | | *t*-tests | |
|---|---|---|---|---|---|---|---|
| | | | | **W** | ***p*-Value** | **t** | ***p*-Value** |
| Male | | | | | | | |
|   AFW | 31 | 0.00 | 0.082 | 0.975 | 0.685 | 0.065 | 0.984 |
|   LFW | 33 | −0.013 | 0.083 | 0.969 | 0.473 | −0.924 | 0.363 |
|   SF | 30 | −0.004 | 0.016 | 0.955 | 0.235 | −1.453 | 0.157 |
| Female | | | | | | | |
|   AFW | 28 | 0.028 | 0.077 | 0.981 | 0.862 | 1.958 | 0.061 |
|   LFW | 40 | 0.007 | 0.023 | 0.956 | 0.120 | 2.015 | 0.051 |

**Note:**
AFW, forewing area; LFW, forewing length; SF, length of the second segment of forceps.

relationship between average rainfall and body length of males (Spearman: $\rho = -0.45$, $p < 0.001$), females (Spearman: $\rho = -0.64$, $p < 0.001$), and both sexes combined (Spearman: $\rho = -0.51$, $p < 0.001$, Fig. 3).

## Fluctuating asymmetry analysis in *F. caribbianus*

A total of 117 *F. caribbianus* (50 females and 67 males) were assessed (in groups of ~30 randomly selected individuals for each test) to determine departure from symmetry. Analyses of trait value distribution satisfied the assumption of normality, so there was no evidence of antisymmetry in any of the characters ($p > 0.05$, Table 1). Also, the *t*-tests revealed that mean (R–L) was not significantly different from zero, which suggests that there was no directional asymmetry ($p > 0.05$, Table 1).

**Table 2 Result of the two-way mixed model ANOVAs performed for each trait with side as a fixed component and individual as random component.**

| Trait | N | Mean error[a] | Mean (mm)[b] | Side | | Individual | | Side × individual | | | $ME_{error}$[d] | FA1 | FA4a | FA10a | ME3 (%) |
|---|---|---|---|---|---|---|---|---|---|---|---|---|---|---|---|
| | | | | df | F | df | F | df | $ME_{SxI}$[c] | F | | | | | |
| Male | | | | | | | | | | | | | | | |
| AFW | 31 | 0.03 | 4.41 | 1 | 18.90* | 30 | 797.40* | 30 | 3.98E−5 | 13.30* | 2.99E−6 | 0.076 | 0.088 | 0.004 | 16.4 |
| LFW | 30 | 0.03 | 3.79 | 1 | 2.05 | 29 | 223.89* | 29 | 3.33E−3 | 5.10* | 6.52E−4 | 0.063 | 0.066 | 0.034 | 19.6 |
| SF | 30 | 0.006 | 0.45 | 1 | 10.70* | 29 | 103.80* | 29 | 4.25E−4 | 7.50* | 5.67E−5 | 0.012 | 0.013 | 0.012 | 13.3 |
| Female | | | | | | | | | | | | | | | |
| AFW | 28 | 0.03 | 5.16 | 1 | 0.01 | 27 | 2163.2* | 27 | 1.38E−5 | 16.9* | 8.27E−7 | 0.065 | 0.062 | 0.002 | 5.9 |
| LFW | 28 | 0.02 | 4.01 | 1 | 8.66* | 27 | 507.30* | 27 | 2.35E−3 | 5.20* | 4.51E−4 | 0.043 | 0.046 | 0.028 | 19.2 |

Notes:
* $p < 0.05$.
[a] Mean of the standard deviation of triplicate measurements on the right and left sides (i.e., indicator of accuracy during photo measurement).
[b] Mean of the right and left side measurements.
[c] Mean squared of the side × individual interaction.
[d] Mean squared of the variance of the repeated measurements.
Indices of asymmetry: FA1, FA4a, FA10a and measurement error as percentage (ME3 as % of $ME_{SxI}$). AFW, forewing area; LFW, forewing length; SF, length of the second segment of forceps.

Two-way mixed model ANOVAs (Table 2) showed that the side × individual interaction was always significant, while global error ($ME_{error}$) was found to be smaller than the error of the interaction ($ME_{SxI}$) for all traits. Measurement error (ME3: $M_{error}$/$ME_{SxI}$ × 100) contributed 5.9–19.6% of the total variance between-sides for each trait, which indicates that ~80% of the measurement variation was reliable. Fluctuating asymmetries for each trait amounted to roughly 1.07–2.67% of the average size of the corresponding trait (FA1/mean). The estimates for FA4a and FA10a varied from 0.013 to 0.088 and from 0.002 to 0.034, respectively.

## DISCUSSION

Our two-year study showed high temporal variability of mayfly benthic biomass, characterized by multiple peaks during the study period. Unexpectedly, these peaks in benthic biomass did not translate into measurable increases in emerging adult biomass, which represented a small fraction of the benthic biomass. Notably, while benthic and emergent biomass were similar during most of the study period, the observed peaks in nymph biomass were not associated to similar peaks in the emergence of adults. Peaks in benthic biomass occurred in different periods of the year, which may suggest that different mechanisms control population dynamics of benthic and emerging mayflies. E/P ratio varied between 38.57% in 2002 and 19.78% in 2003, and was higher than reports in previous studies for emerging aquatic insects in temperate regions. We did not observe a relationship between hydrology and the biomass of benthic and emergent mayflies, so our expectation that mayfly biomass would peak in the dry season of La Selva was not supported by our findings. Looking more closely at *F. caribbianus*, the most abundant mayfly collected in emergence traps, we found no evidence of antisymmetry or directional symmetry, but we did find slight deviations from symmetry that were larger than measurement errors, suggesting alterations in bilateral morphology of *F. caribbianus*.

We also found a strong negative relationship between body length and precipitation variability at La Selva. This relationship was consistently significant for males, females, and both sexes combined.

Our E/P estimation indicated that less than 40% (38.57% and 19.78% for 2002 and 2003, respectively) of nymph secondary production was exported to the terrestrial ecosystem. This potentially low value may have negative consequences on the riparian food web, as emerging insects represent an important source of energy and nutrients in La Selva streams (N-flux: 0.40–1.25 mg N m$^{-2}$ d$^{-1}$, *Small et al., 2013a*). Surprisingly, our E/P estimates were equal to, or exceeded, the values previously reported for single groups of aquatic insects, such as 19.3% for *Hydropsyche angustipennis* and 27% for *H. pellucidula* (Trichoptera) in a Northern German lowland stream (*Poepperl, 2000a*), and 16% for a limnephilid (Trichoptera) in an intermittent wetland (*Whiles, Goldowitz & Charlton, 1999*). Our E/P estimates were also high when compared to entire insect assemblages, such as 16.6% in desert streams (*Jackson & Fisher, 1986*) and 18.3% in a northern German stream (*Poepperl, 2000b*). Our E/P estimates even exceed the biomass export of aquatic vertebrates, such as salamanders (10%, *Regester, Lips & Whiles, 2006*). These elevated E/P ratios may be explained by the warm water temperatures (range: 21.4–27.2 °C, *Gutiérrez-Fonseca, Ramírez & Pringle, 2018*), and high biomass turnover rates of fast-growing mayflies (P/B= 13.63 y$^{-1}$, *Ramírez & Pringle, 1998*) that are characteristic of many tropical streams.

Unlike the large fluctuations in emergence patterns observed in other studies (*Masteller, 1993*; *Pescador, Masteller & Buzby, 1993*; *Castro-Rebolledo & Donato-Rondon, 2015*; *Yuen & Dudgeon, 2016*), which provide support for seasonality of many tropical aquatic insects, the lack of abrupt peaks in mayfly emergence found in our study suggests that mayfly emergence was aseasonal. Fish predation on newly emerged adults may be a possible explanation for the constant biomass of emerging mayflies in our study. Our focal stream is inhabited by the insectivorous poeciliid, *P. annectens*, which is abundant (4–14 individuals/m$^2$, *Small et al., 2013b*) and could have negatively affected mayfly emergence. Previous studies have shown a reduction of total emergence biomass of aquatic insects by 62 ± 8% (*Merkley, Rader & Schaalje, 2015*) and 65% (*Warmbold & Wesner, 2018*) in mesocosms with a similar fish density (11.4 fish/m$^2$ and 7.8 fish/m$^2$, respectively) to our study stream. Field studies have also found that fish can regulate the timing and duration of aquatic insect emergence (*Moore & Schindler, 2010*). Therefore, it is not surprising for biotic control to be an important factor controlling insect emergence in our tropical lowland streams, which harbor high fish diversity.

Hydrology has been recognized as a key factor controlling macroinvertebrate assemblages in tropical streams (*Flecker & Feifarek, 1994*; *Ramírez & Pringle, 1998*; *Molineri, 2010*), with peaks in biomass expected during the dry season and high mortality during the rainy season. However, we observed peaks in benthic biomass during both the dry and rainy seasons in La Selva. Peaks during the rainy season might be caused by microdistributional changes in macroinvertebrates driven by high floods (*Lancaster & Hildrew, 1993*; *Lancaster, 1999*). For instance, mayflies have been observed to make small-scale refuge-seeking movements between substrate layers during simulated floods

(*Holomuzki & Biggs, 2000*). This type of mechanisms could allow for high mayfly survival in streams such as Carapa, where the deep subsurface layers may provide shelter and protection for insects year round, as proposed for similar streams (*Holomuzki & Biggs, 2000*).

Our results show a strong influence of precipitation on the total body length of *F. caribbianus*. Large individuals were negatively affected by precipitation, while small-sized mayflies persisted during high rainfall events (Fig. 3). In tropical streams, two mechanisms have been identified as potential ways in which insects respond to catastrophic floods. The first mechanism proposes that small-sized individuals have better chances of surviving floods by finding refuge in interstitial spaces (*Townsend, Dolédec & Scarsbrook, 1997*; *Segura, Siqueira & Fonseca-Gessner, 2013*). The second hypothesis suggests that some aquatic insects (e.g., Odonata: Polythoridae) emerge, copulate and oviposit at the onset of the rainy season, so that only small larvae (which can be protected by large logs and rocks) are present during periods of frequent floods (*Pritchard, 1996*). Since mayflies in La Selva are multivoltine (*Ramírez & Pringle, 1998*), we find more support for the first hypothesis through this study.

Asymmetrical traits have been used successfully as an early warning biomarker related to developmental stress (*Graham et al., 2010*). Common stress sources that cause fluctuating asymmetries in aquatic insects include water quality (*Bonada & Williams, 2002*), insecticide (*Mpho, Holloway & Callaghan, 2001*), experimental food deprivation (*Nosil & Reimchen, 2007*) and changes in the physical structure of riparian vegetation (*Pinto et al., 2012*). We found slight variations from perfect symmetry in the wings and forceps of *F. caribbianus*, which suggests some level of developmental instability of the nymph. However, given the limited number of mayflies per period, we were unable to perform analyses under different levels of environmental stressors (e.g., low, medium and high hydrology). Therefore, although we found slight random deviations from symmetry, they were not distinguishable from developmental noise (i.e., random variation from symmetry caused by metabolic rates, concentrations of regulatory molecules, diffusion, or thermal noise) without further study (*Palmer & Strobeck, 2003*).

Future climate change scenarios predict an increase in hydrological extreme events for many regions (*Christensen et al., 2013*). Extreme precipitation events are expected to increase in tropical regions (*O'Gorman & Schneider, 2009*), with potential negative effects on aquatic biota and aquatic-terrestrial linkages. Increases in heavy precipitation events have already been observed in the Caribbean slope of Costa Rica during the last decades (*Aguilar et al., 2005*; *Rapp et al., 2014*; *Sánchez-Murillo et al., 2017*), where climate projections suggest an increase in mean annual precipitation of between 10% and 50% (*Alvarado et al., 2012*). Therefore, large hydrological variability can threaten the fitness of mayfly populations in La Selva, as well as in other tropical regions.

## CONCLUSIONS

Contrary to our expectations and patterns shown in literature, we found a lack of seasonality in benthic biomass. Adult biomass was unrelated to peaks in benthic biomass,

which makes us wonder what is controlling adult biomass export in these systems if not hydrology (e.g., fish predation). Based on our E/P ratios, Neotropical streams can provide larger subsidies to adjacent terrestrial ecosystems than their counterparts in temperate regions. Departures from perfect symmetry were evident in all the evaluated traits, which suggests developmental instability of mayflies. Body length proved to be a better indicator of environmental stress, which could have severe associated costs for mayfly fitness in ecosystems with high temporal variation. Further research could quantify effects of body length reduction in mayfly fitness, energy and nutrient export to riparian food webs, as well as the role of biotic control on mayfly biomass in tropical lowland streams.

## ACKNOWLEDGEMENTS

We are grateful to Minor Hidalgo for his invaluable assistance in collecting benthic and adult samples. Thanks also to Eduardo Domínguez for identifying the *Farrodes* species, José A. Sánchez-Ruiz for his assistance in the laboratory, and Aura M. Alonso-Rodríguez for revising earlier drafts of the manuscript. We thank the Organization for Tropical Studies and the staff of La Selva Biological Station for their help in facilitating this research.

### Funding

This work was supported by the National Science Foundation, under Grant DEB-1938843. The funders had no role in study design, data collection and analysis, decision to publish, or preparation of the manuscript.

### Grant Disclosures

The following grant information was disclosed by the authors:
National Science Foundation: DEB-1938843.

### Competing Interests

The authors declare that they have no competing interests.

### Author Contributions

- Pablo E. Gutiérrez-Fonseca conceived and designed the experiments, analyzed the data, prepared figures and/or tables, authored or reviewed drafts of the paper, and approved the final draft.
- Alonso Ramírez conceived and designed the experiments, performed the experiments, analyzed the data, prepared figures and/or tables, authored or reviewed drafts of the paper, and approved the final draft.

### Data Availability

The code is available at GitHub: https://github.com/PEGutierrezF/mayfly_morphometry.

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
