# Peer review of "Mayfly emergence production and body length response to hydrology in a tropical lowland stream"

_PeerJ, doi:10.7717/peerj.9883_

## Round 0.1 · original submission · Major Revisions

Dear Drs. Gutiérrez-Fonseca and Ramírez:

Thanks for submitting your manuscript to PeerJ. I have now received three independent reviews of your work, and as you will see, one reviewer recommended rejection, while another suggested a major revision (with many suggested changes). I am affording you the option of revising your manuscript according to all three reviews but understand that your resubmission may be sent to at least one new reviewer for a fresh assessment (unless the reviewer recommending rejection is willing to re-review).

The reviewers raised many concerns about the manuscript. Please address all of these in your rebuttal letter. I especially would like to see your response to Reviewer 3’s concern over hydrology as a stressor for the benthic communities.

Good luck with your revision,

-joe

Reviewer 1 ·

Basic reporting

See "General comments for the author"

Experimental design

See "General comments for the author"

Validity of the findings

See "General comments for the author"

Additional comments

Dear Editor,

I have read and reviewed the paper entitled “Hydrology influences body length, but not benthic and emergent biomass of Ephemeroptera in a tropical lowland stream” by Gutiérrez-Fonseca & Ramírez, which has been submitted to PeerJ for your consideration (manuscript number 47058).

The authors study the traits declared in the title for a specific mayfly species (Ephemeroptera: Leptophlebiidae) called Farrodes caribbianus. The interannual variability in precipitation that characterizes the tropical stream they study (Carapa stream) is presented as a stressor, yet precipitation does not seem to influence asymetry which is known to respond to environmental stress. In contrast, body length (but not biomass) is affected by hydrology. The authors use monthly data for years 2002 and 2003 to make their case.

The Introduction section is well written and easy to follow. I only made few specific comments, most of them really minor. Nevertheless, predictions are described as hypotheses, although this could easily be fixed with some rephrasing. Also, I think adding a couple of references (particularly recent references, if possible) at specific places (see comments below) would be beneficial. Methods are also well written and very well explained. Only some minor suggestions apply here as well. Results indicate absence of effect of environmental variables on biomass or asymmetry, but a significant effect on body length.

One drawback of this study is that it might be problematic to have a short time series (two years) to prove any influence of interannual variability on biomass and asymetry, in my opinion. On the other hand, it is true that even with a short time series the authors were able to find a clear (inverse) relation between body length and precipitation. But the interpretation of results (i.e. the Discussion section) is still not convincing, in my view.

For example, with respect to biomass, it is stated that high floods may be responsible for microdistributional changes which in turn may result in changes in biomass. However, this was never proven with the precipitation data that the authors have at their disposal. With respect to asymmetry, no environmental variable was significant in their model. The explanation given in this case relies on a possible trade-off between body size and asymmetry, in the sense that the priority would be to preserve symmetry and thus function, at the expenses of body size variation. It can be reasonable to argue that body size changes are less critical to fitness than deviations from symmetry might be. However, the connection between the two in more mechanistic terms is unclear to me and might not be so straightforward. A more detailed explanation is needed, in my opinion. Also, I wonder if it could be possible that other environmental variables that weren’t measured are actually responsible for the observed variability in body size. Also in relation to body length, it isn’t very clear why body length (and not biomass) is responding to precipitation.

Overall, I don’t see this manuscript ready for publication in its present form, but maybe the authors are able to provide a more convincing case after major revision. This should include other forms of measuring precipitation at appropriate scales of analysis. The Discussion section needs to improve substantially so as to address the comments mentioned above (see also the specific comments provided below).

Sincerely,
the reviewer.

Specific comments:

Abstract:

Line 38: “Our multi-year study”. However, this study only covers two years: 2002 and 2003.

Lines 39-40: “unrelated to hydrological fluctuations, but potentially related to trophic interactions (e.g. fish predation)”. However, the absence of relation to hydrogical factors could change if precipitation were estimated otherwise (i.e. at other scales of analysis). On the other hand, the relation to “trophic interactions (e.g. fish predation)” has not been proven with the data available. Overall, the absence of relation to hydrological fluctuations could result from considering precipitation at the wrong scale of analysis (e.g. single events may have more influence than average rainfall) and the potential influence of fish predations is weakly supported by results. Both issues undermine the conclusions drawn.

Introduction:

Lines 49-51: “In lotic environments, much attention is given to understand the impact of environmental variables on aquatic organisms at multiple levels, from individuals to the community”. However, this reads like if this were a particular feature of studies focusing on lotic environments. I would rephrase this to highlight that lotic environments are not an exception in this regard.

Line 53: I think adding one more recent reference here would be a good idea.

Lines 76-78: “The first two cases of asymmetry are related to genetic changes, while fluctuating asymmetry (FA) is caused by environmental stressors”. I think there should be a reference here.

Line 77: “fluctuating asymmetry (FA)”. This abbreviation “FA” is seldom used in the text, tables and figures. Therefore, I would eliminate “(FA)” from here.

Lines 93-95: “we used data collected during two years (2002-2003) to determine temporal patterns of emerged Ephemeroptera and the benthic standing-stock biomass”. I think the temporal span (2002-2003) might be too short to draw conclusions about the effect of interannual environmental variability on biomass and asymetry.

Line 98: “(Leptophlebiidae, Domínguez, 1999) adults”. I would eliminate “Leptophlebiidae” from here because it reads strange together with a reference. And, also “adults” could be placed somewhere else to facilitate reading. Perhaps the entire sentence could be rephrased as follows: “we examined departures from perfect symetry in adults of the Leptophlebiidae Farrodes carribamus (Traver) comb. nov. (Domínguez, 1999), ... ”.

Lines 102-104: “We hypothesized a peak in benthic and emergent biomass during the low rainfall season, due to a decrease in the risk of mortality from drag during floods”. I think the “hypothesis” here is the “decrease in the risk of mortality from drag during floods” while the “peak in benthic and emergent biomass during the low rainfall season” is the “prediction” that results from this hypothesis. I would rephrase this so as to be more accurate in this regard.

Lines 104-106: “We also hypothesized that potential changes in adult symmetry and body length would reflect nymph development instability due to exposure to environmental stressors during their lifespan, and their ability to buffer environmental disturbances”. I think here the “hypothesis” is that “exposure to environmental stressors during their lifespan relate to nymph development instability” while the “prediction” resulting from this hypothesis is a “change in the adult symmetry and body size”. I would rephrase this a little bit as well.

Methods:

Line 113: “1563ha”. Space needed.

Line 115-116: “Long-term average annual precipitation (1963-2000) is 4314 mm, ranging from 2809 mm in 1995 to 6164 mm in 1970”. Can this be updated to present dates (e.g. from 1963 to 2019)?

Line 120: “100m. Space needed.

Line 136: “Three core samples (0.006 m2 each)”.

Line 151: “AFDM”. Please develop the meaning of this for non-specialized readers.

Line 171: “abbreviated to BL”. However, this abbreviation is never used in the text, nor in Tables or Figures. Therefore, I would keep using “body length” all the time and eliminate this explanation here.

Lines 176-177: “all measurements were taken three times from the same image at a random order”. I didn’t get this. So, the same image is used to perform three measurements at a random order. Does this mean that many different images were displayed at a random order so that in the end all images were shown three times? I think this needs some more detailed explanation.

Lines 183-186: “biomass better reflects potential changes in mass production”. This is circular. I would eliminate this. In fact, I would eliminate the entire paragraph (Lines 183-186) because it doesn’t add anything relevant for the purpose of this study.

Line 206: I would replace “FA” by “fluctuating asymmetry”. The definition of “FA” was made in the Introduction section (Line 77) but seldom used elsewhere in the text. I didn’t remember what it meant when I found it here again for the first time.

Line 208: I would replace “FA” by “fluctuating asymmetry” also here.

Results:

Line 222: “the 2002 and 2003”. I would eliminate “the”.

Lines 229-230: “The AIC analysis used to determine the relative importance of environmental variables on the biomass of emerging adults showed no support for any model”. Maybe other environmental variables not registered in this study are responsible for biomass variability?

Lines 230-231: “This trend was consistent for mayflies collected during 2002 and 2003”. I don’t think this is necessary because this study is about mayflies, and about the years 2002 and 2003. This was made clear in the Introduction and there is no need to repeat it here. In fact, in may be confusing as it reads like if there were other invertebrates or other years available.

Line 241: I would replace “FA” by “fluctuating asymmetry”.

Lines 244 and 246: “Appendix 1”. Where is this Appendix? I don’t find it in this pdf which is the only file associted to this submission that I can find online. See also comments for Table 2.

Line 247: I would replace “FA” by “fluctuating asymmetry”.

Line 251: I would replace “FA” by “fluctuating asymmetry”.

Discussion:

Line 256: “Our 2-y study”. I would write it as “Our two-year study” as it reads better.

Lines 260-261: “our hypothesis that mayfly biomass would peak in the dry season of La Selva was not supported by our findings”. This is the “prediction” that was not observed. The hypothesis not supported is the “decrease in the risk of mortality from drag during floods”

Lines 264-266: “we found a strong negative relationship between body length and precipitation variability at La Selva. This relationship was consistently significant for males, females, and both sexes combined”. How do the authors interpret this finding? Is it related to e.g. resource availability which would decrease with precipitation? Or maybe related to the necessity of synchrony in emergence, which perhaps obligates nymphs to stay longer (or shorter) periods under water and therefore grow more (or less)? Do they have an explanation for this? I think the explanation given in Lines 310-317 suits well right here.

Lines 274-275: “these studies measured productivity (g/m2/y), while we reported standing-stock biomass (g/m2)”. Can the authors develop this a little bit more? Are these measurements comparable? How?

Lines 282-285: “the lack of abrupt peaks in mayfly emergence found in our study suggests that mayfly emergence is aseasonal. A possible factor explaining the stability in adult emergence is fish predation on newly emerged adults”. However, the authors also mention that this isn’t happening in other studies on tropical streams. Can they provide data about fish abundance or fish predation rate comparing their study to previous studies? If not, this explanation seems to have weak or no support.

Lines 285-286: “Wesner (2016) showed that fish predation decreases the number of insects reaching riparian ecosystems by 40% on average”. Would this 40% be enough to produce aseasonality? What this number would be in this study? Do the authors have an estimate?

Lines 287-289: “Our focal stream is inhabited by the insectivorous poeciliid, Priapicthys annectens, which is abundant (4-14 individuals/m2, Small et al. 2013) and could have negatively affected mayfly emergence”. What about the other studies? Are predation rates or fish abundance lower in those streams having seasonality in emergence?

Lines 292-294: “However, observed peaks in benthic biomass occurred in both dry and rainy seasons in La Selva. This may be due to high floods during the rainy season eliciting microdistributional changes in macroinvertebrates”. However, biomass wasn’t related to any of the environmental variables considered (Lines 229-230), which included “montly precipitation” (Line 165). Thus, I think the results don’t support this interpretation. Maybe monthly precipitation is not accurate enough and other measurements of precipitation may support better this view?

Lines 310-317: This paragraph would be better placed right after Lines 264-266 where an explanation was missing.

Line 300: I would say “In contrast” (or something similar) instead of the more neutral term “Meanwhile”.

Conclusions:

Lines 332-333: “Body length, and not bilateral asymmetry, proved to be a better indicator of environmental stress”. However, what type of stress is normally associated to asymmetry in other studies? Is it anomalous precipitation or is it pollution? In fact, the second hypothesis (see comments for Lines 104-106 for hypothesis definition) was not very well discussed, in the sense that it isn’t that clear whether or not this hypothesis is rejected (as it is done with the first hypothesis, see comments for Lines 260-261). It is true that a trade-off between body length and asymetries is mentioned, rather than truly discussed in the context of the environmental variables in this study (Lines 315-317). In any case, maybe it would be helpful to develop a little bit in the Discussion section which types of environmental stressors are more likely to actually induce asymmetry.

References:

I wouldn’t use DOI numbers as many reference don’t have and so it feels weird (i.e. non-standard) that some have (the more recent) and some don’t (the older). I would use DOI only in case the manuscript has been accepted for publication (or it is already published as EarlyView) but doesn’t have volume and pages assigned yet. However, this is just a personal view and in fact I didn’t even check the journal instructions with regard to this specific detail. But in my opinion DOI numbers make the reference list less neat and do not contribute to clarity in this regard.

Tables:

Table 2: I would specify also here that “Side” is the fixed component and “Individual” the random component of this ANOVA analysis. In this table the authors provide p-values (for the variable “individual” and the categoy “male”) in the form of “p>0.001”. I assume that the correct Table 1 is the one uploaded as “Supplementary Materials”. In fact, this is the only information I was able to find in the “Supplementary Materials”. I would therefore replace Table 1 in the main text by Table 1 in the Supp Info. Also, I was expecting to find “Appendix 1” as part of the Supplementary Materials, but I didn’t find it there nor elsewhere.

Supplementary Materials:

See comments for Table 2.

Sincerely,
the reviewer.

Reviewer 2 ·

Basic reporting

Please see below

Experimental design

Please see below

Validity of the findings

Please see below

Additional comments

General

This manuscript reports the results of a study of mayfly nymphs and adults in a small first-order stream on the Caribbean slope of Costa Rica. Biomass of nymphs was assessed with a core sampler and biomass of adults with a 2m2 emergence trap monthly over 2 years (nymphs) and weekly (adults). At the same time, 8 environmental variables were recorded monthly (nitrogen as nitrate and ammonium, soluble reactive phosphorous, temperature, pH, conductivity, discharge, precipitation). The hypothesis addressed was that there would be a peak in benthic and emergent biomass during the low rainfall season due to reduced mortality from drag during floods. To test this, precipitation was measured rather than flows within the stream because previous research has “demonstrated their influence on LSBS streams”. Secondly, the authors hypothesized that hydrological stresses suffered by nymphs of the most abundant mayfly, Farrodes caribbianus, would produce bilateral asymmetry in adults and changes in body length. Benthic biomass was variable but was not reflected in emergence biomass which was relatively constant. No evidence of bilateral asymmetry was observed but adult body length was negatively correlated with average precipitation in the preceding 159 days (an estimate of the duration of the nymph stage). Overall the questions addressed by this study are interesting, the study design is suitable for addressing the questions and the results appear to be appropriately analyzed and interpreted. Data are clearly presented in tables and figures and the writing is clear. I recommend that this paper be published after consideration of the questions and suggestions below.

Details
- nymph biomass was estimated from monthly triplicate core samples in leaf-covered runs in a 100m reach of the small (1m wide, 25 cm deep) Carapa Stream. Can you say anything about spatial variance within this reach so the reader may understand how representative each monthly sampling was and the likelihood that the data in Figure 1 represent temporal variance, as suggested, rather than spatial variance or some combination of spatial and temporal variance?
- when a mayfly nymph emerges from the substrate and leaves the water, where would I find that individual relative to where it had lived in the substrate as a nymph? In a fast flowing stream I would expect it to leave the water some distance downstream from where the nymph had been in the substrate because the stream would carry it before it had a chance to reach the surface and fly free. I don’t have a feel for this in your stream because I don’t know how fast the flow is or how long the nymph would be in the water column before it is able to break the surface and fly free. The relevance of this is that, if you want to relate emergent adults to nymphs, you would want to place your emergence trap some distance downstream of the patch of stream in which you sampled nymphs wouldn’t you? Can you tell us whether you took this into consideration in your sampling design?
- It would be helpful to describe the relationship between rainfall and flow discharge or velocity in the stream to bolster the argument that precipitation relates to increased forces being exerted on nymphs in the substrate. How strong is the correlation between precipitation and current speed and what temporal lag is there between a rain event and increased flow in your creek?
L70-71 – The logic of this sentence is not clear. Do you mean that bilateral symmetry is known to be best in some way (what way? Reproductive success?) and so any deviation from bilateral symmetry puts the animal at some disadvantage? Given that environmental stresses can produce deviations from bilateral symmetry, measurements of bilateral symmetry are accepted as an index of environmentally relevant perturbation? You go on to make that argument for body length but not explicit for asymmetry.
L76-78 – include references to support this statement
L79 – add “s” to alter
L85 – no need for hyphen between long and periods. Add “days” after 165
L86 – add “d” to compare
L97 – add “98” after nov. (?)
L104 - define drag – just force of water movement?
L125 – pH of 3.62 is very low – why did the El Nino event of 1997-98 produce this effect?
L151 – define AFDM (presumably Ash Free Dry Mass?)
L170 – change were to was to agree with a suite
L176 – did the same person make all measures? How close were the measures? Can you provide a measure of variance?
L201 – data were (rather than was)
L204 – add s to model (?)
L233 – tell us why you used a Spearman’s rather than Pearson’s correlation
Figure 2 – I presume the grey band is a 95% Confidence Interval? Explain in figure caption
Table 2 – define all acronyms in Table caption so table can stand alone (i.e., be understood without reference to text)
- should all > symbols actually be <?
- Provide degrees of freedom for tests

Reviewer 3 ·

Basic reporting

The study presents a substantial amount of data that can potentially provide interesting results if the study would have been framed in a stronger conceptual framework. Unfortunately, the presented manuscript fails to convince and suffers from the lack of a strong theoretical background. Some statements by the authors are not supported by any literature. For example, in lines 71 to 78, genetic causes are presumed to lead to directional symmetry and antisymmetry while fluctuating asymmetry is presumed to be caused by environmental stressors. Would that refer to phenotypic plasticity? Stronger arguments are needed.

Experimental design

My main concern is how hydrology was handled and considered as a stressor for the benthic communities. The variable chosen by the authors as a proxy for hydrology was average precipitation, and later on, they expand in the discussion with direct links to flood as the main cause for change in F. caribbianus total body length. While the relationship between precipitation and body length was significant, it is very important to remind the reader that this decrease in body size was established only on 117 individuals. A very low number of individuals to be analyzed considering this taxon to be the most abundant mayflies in the study site. Also, precipitation can be highly correlated to changes in air temperatures. Did the authors examine such correlation? How can the authors be certain that precipitation (then flow), is behind the change of total body length? wouldn’t it be the temperature?

Validity of the findings

The findings are partially speculative and fail to establish a clear causality relationship between hydrology and body length.

Additional comments

Here are few more minor mistakes that need to be addressed, even if at that point I did not thoroughly examine the manuscript for grammatical and typographical mistakes.

Lines 187 to 190: More information on the model selection needs to be provided in the manuscript. Referring to published literature is not sufficient.

Line 221 to 223: Emergence represents 24.4 to 14.9% of the benthic biomass? please clarify.

Line 85: Add “days”, after range from 26 to 165 …
Line 123: Silt and clay are fine sediments not only sediments.

Table 2: When using codes in the table please introduce them in the caption, and no need for a separate table for that.
Figure 2: Need to specify in the caption that this data represents only Farrodes caribbianus.

---

## Round 0.2 · Minor Revisions

Dear Drs. Gutiérrez-Fonseca and Ramírez:

Thanks for revising your manuscript. The reviewer is very satisfied with your revision (as am I). Great! However, there are a few minor things to address. Please address these ASAP so we may move towards acceptance of your work.

Best,

-joe

Reviewer 2 ·

Basic reporting

The authors' revision of their manuscript addressed all of my concerns (and from my reading most concerns raised my the other two reviewers) adequately with the exception of three:
1. Nymph biomass was estimated from monthly triplicate core samples in leaf-covered runs in a 100m reach of the small (1m wide, 25 cm deep) Carapa Stream. Can you say anything about spatial variance within this reach so the reader may understand how representative each monthly sampling was and the likelihood that the data in Figure 1 represent temporal variance, as suggested, rather than spatial variance or some combination of spatial and temporal variance?
2. did the same person make all measures? How close were the measures? Can you provide a measure of variance
3. data were (rather than was)
I consider these sufficiently minor that I am happy to recommend acceptance of the manuscript at this point.

Experimental design

as above

Validity of the findings

as above

---

## Round 0.3 · accepted · Accept

Dear Drs. Gutiérrez-Fonseca and Ramírez:

Thanks for revising your manuscript based on the concerns raised by the reviewers. I now believe that your manuscript is suitable for publication. Congratulations! I look forward to seeing this work in print, and I anticipate it being an important resource for groups studying mayfly biology and ecology. Thanks again for choosing PeerJ to publish such important work.

Best,

-joe